# OpenReview forum: "Self-Taught Optimizer (STOP): Recursively Self-Improving Code Generation"
_colmweb.org/COLM/2024/Conference — COLM_

### Official Review · Reviewer_wTSv · 2024-05-10

**Rating:** 9
**Confidence:** 3
**Ethics Flag:** 1

**Summary:**

The paper is broadly about prompt optimization, specifically about using the LLM itself to iteratively refine and improve an initial seed prompt. The paper asserts that the various LLM prompt engineering techniques and methods are versions of the same general optimization problem, provides a theoretical formulation of this problem, then proposes the novel STOP algorithm as a way to leverage the LLM itself to conduct the optimization method search. The paper demonstrates this on the task of generating Python code implementations of solutions to lesser-known Computer Science problems. STOP is evaluated primarily on a checkpoint of GPT-4, though some experiments were conducted on checkpoints of GPT-3.5 turbo and one open source model, Mixtral-8x7B. Relative improvements are highlighted in the results, though the discussion primarily focuses on the nature and generalizability of the optimization methods produced by the algorithm. Special attention is given to instances where generated code produced by the LLM appears to have bypassed safeguards.

**Questions To Authors:**

What was the actual usage in terms of number of calls / tokens consumed / cost per iteration or full run? While LLM consumption costs may not be anywhere near a limiting factor as actual LLM training, it would still be nice to have even a rough estimate of the usage and costs to conduct these experiments from a reproducibility and practicality standpoint. I did not carefully read the entire Appendix, so apologies if this information was included there.

**Reasons To Accept:**

There are several reasons to accept this paper. It is clearly written and motivated, and effort is spent to frame the work properly in context of the broader literature. The idea of using an LLM to conduct an optimization search through iterative code generation is intriguing, and I believe it has broader implications for researches and practitioners wishing to achieve certain outcomes from an LLM without having access or ability to fine-tune. The awareness of the limitations of the presented work and engaging with the broader topic of AI safety are both appreciated. Reproducibility concerns are addressed in the appendix, including that code is available via GitHub.

**Reasons To Reject:**

I do not see any specific reason to reject this paper. One relative weakness is a bit of disconnect between the initial general claims of the STOP algorithm and the practicality of it as demonstrated i.e. limited to code generation, specifically of individual algorithms to problems with well-defined utility functions that are easy to evaluate and describe. There is nothing wrong with this at all, but it's somewhat hard to see how we get from the reality of the experiments to the more sweeping general claims. Another minor issue is the occasional over-reliance on the Appendix for examples and details.

---

> ### Author Rebuttal · Authors · 2024-05-30
>
> Thank you so much for the great points and supportive comments! If there’s any claim in particular that you’d like us to tone down, please let us know and we’d be happy to revise it. It would be interesting to also apply this in a text-only setting and help with the generality of the claims – however, we imagine an optimizer that improves language inputs generally (e.g. rewriting text to sound better) may be less likely to be a good code optimizer, so this would be a nontrivial but important follow-up. We’ll mention this in our revised version. And leaving out the actual total budget was an oversight, thank you for pointing it out – roughly 3 thousand GPT4 calls were used per iteration per run.

---

> > ### Comment · Reviewer_wTSv · 2024-06-06
> >
> > Thank you for the response and additional details. Upon re-review, I am unable to point to any claim specifically that I would alter, as there do appear to be sufficient qualifications about the scope of the work. It may be that I glossed over them on first read through.

---

### Official Review · Reviewer_vbkp · 2024-05-13

**Rating:** 6
**Confidence:** 4
**Ethics Flag:** 1

**Summary:**

The authors propose to take advantage of scaffolding programs with multiple calls to a LLM in an iterative self-improvement loop. In this framework, an “Improver” function is iterated on with the help of a code solution scoring mechanism based on the average score over a set of downstream tasks. The authors also look into the type of self-improvement strategies proposed by the fixed language model, such as genetic algorithms, simulated-annealing based search.

**Questions To Authors:**

I suggest moving the function definition of $\tilde{u}$ in Algorithm 1 above the main recursive loop to make it easier for the reader to understand where $\tilde{u}$ is coming from.


Have you looked at how big each step of optimizing the improver $I$ is? Do you observe any big shift in the improver's scaffolding program at each iteration? If so, have you looked at perhaps implementing some sort of rolling average to keep the updates small enough?

At each step of applying the improver, there could be a big variance between the scores of extracted new solutions, and perhaps taking the “max” to be representative of that step is not the best. Have you looked at this potential issue and perhaps observed the use of an average of utility scores over new solutions as the representative score in updated improvers?

What is the prompt that meta optimizes the improver? And is there any filtering or check on an updated improver? (e.g. the code should be executable, a call to a LLM should exist, etc)

**Reasons To Accept:**

The concept of a recursively self-improving system is interesting, and the proposed way of implementing it (dubbed STOP by the authors) seems straightforward and simple.

The use of a utility score function for the generated code to evaluate and provide feedback for the meta-optimization of the improver is a nice idea.

The paper provides good insights into the self-improving behaviours of the framework.

**Reasons To Reject:**

Novelty of the idea: the concept of the STOP algorithm seems very close to the concept of APE (Zhou et al., 2023) optimization algorithm where a LLM is queried for new prompt candidates which are scored and the one with maximum score is chosen.

A limitation of this type of self improvement is that the meta-optimizer (what improves the improver) is fixed and could potentially limit the progress of the improver itself.

The proposed method seems to not work with GPT 3.5 turbo and Mixtral which suggest that the method or the chosen utility function is not robust or a careful choice of seed improver or utility function is required.

I think there aren’t enough baselines in the paper. It would be fair to have APE and another iterative prompt optimization method, such as DLN (Sordoni et al 2023) as baselines for the downstream tasks with the same LLM as base.

---

> ### Author Rebuttal · Authors · 2024-05-27
>
> Thank you for the excellent questions! We wanted to start by clarifying one key detail, which we hope clarifies a few questions at once:
> > What is the prompt that meta optimizes the improver?
>
> We use the improver as its own improver, where instead of passing the solution to some downstream task, we pass the improver code itself and use the meta-utility as the objective. So there's actually no separate prompt: the improver prompt is also the improver improver prompt.
>
> > A limitation of this type of self improvement is that the meta-optimizer (what improves the improver) is fixed and could potentially limit the progress of the improver itself.
>
> After each iteration, we use the improved improver from the previous iteration to improve itself, which isn't fixed. This is one of the key ideas of the paper, but the levels of meta-ness can make it easier to miss.
>
> > Novelty of the idea: the concept of the STOP algorithm seems very close to the concept of APE (Zhou et al., 2023) optimization algorithm where a LLM is queried for new prompt candidates which are scored and the one with maximum score is chosen.
>
> Notably, one of the key differences from APE is that the improver is improving itself. You can see the APE as one specific kind of improver scaffold that the model might use, and indeed, many improvers it proposes are similar to APE, but the meta-optimization is the focus of this paper.
>
> > At each step of applying the improver, there could be a big variance between the scores of extracted new solutions, and perhaps taking the “max” to be representative of that step is not the best. Have you looked at this potential issue and perhaps observed the use of an average of utility scores over new solutions as the representative score in updated improvers?
>
> Yep! We define the meta-utility according to the expectation of downstream performances. For each improver, naturally we cannot control the criteria by which the model selects improvements (and indeed, some of the model-proposed solutions use upper-confidence bound / bandit approaches given this uncertainty).
>
> > I suggest moving the function definition of $\hat{u}$ in Algorithm 1 above the main recursive loop to make it easier for the reader to understand where $\hat{u}$ is coming from.
>
> Good idea! We'll change this.

---

> > ### Comment · Reviewer_vbkp · 2024-06-04
> >
> > Thank you so much for the response and clarifications.
> >
> > I have some follow-up questions.
> >
> > In the improver improving process, during meta-utility calculation, do you only apply the current improver only once on the initial solution? I'm curious to know if the meta-utility function could be improved if the improver was applied either a fixed number of times on an initial solution or if it was applied on a couple of initial solutions per task?
> > How noisy is the meta-utility function given that it is the only (and most important) signal to improve the improver with?
> >
> > Is there a guarantee that using $improver_t$ is better than ${improver}_{t-1}$ at improving an "improver" ? From the meta-utility function we know that it would be better at improving initial solutions for a set of tasks, but is improving an improver the same as improving an initial solution of a task (which I believe is of a different nature)?

---

> > > ### Author Response · Authors · 2024-06-06
> > >
> > > Great questions!
> > >
> > > We define the meta-utility (as in Algo 1) as the average downstream task utility obtained when the improver is applied once to D independently sampled downstream task problem instances. For our main experiments we sampled 5 identical instances of the improvement task (see Sec 5.1). In practice, the meta-utility is stochastic and a few proposed model solutions depend on this (like a bandit that aimed to select the best improvement). And as you point out, there's no guarantee that a self-improved improver is actually better than the previous in terms of meta-utility, and for the weaker models we see this in practice. This also makes the task of finding an improver that works on a purely language-based downstream task (rather than just code) more tricky.
> > >
> > > Thank you again!

---

### Official Review · Reviewer_NEZD · 2024-05-13

**Rating:** 5
**Confidence:** 4
**Ethics Flag:** 1

**Summary:**

A meta-self-improvement procedure is described for code-generating language models. The procedure is seeded with a basic program, with access to a LLM API, that proposes revisions to a given program and selects the best one. The program is then called to revise *the revision program itself* to obtain an improved revision program, and these steps are iterated. It is shown that interesting algorithms resembling beam search, genetic search, etc. are discovered by this procedure. These algorithms produce better programs for combinatorial tasks that also transfer well across tasks. Some possible unsafe behaviours (e.g., hacking the reward mechanism by exploiting the lack of a tensor shape check) are also illustrated and discussed.

**Questions To Authors:**

Please see above.

Meta-note: I am likely to increase the rating if my questions are addressed.

**Reasons To Accept:**

- The paper discovers a very interesting emergent phenomenon in LLMs.
  - It can have a significant impact if it can be generalized to different problems and applied to more accessible LMs.
  - It additionally helps to understand potential risks of autonomous AI systems with self-improvement goals. The attention to studying the possible risks of the proposed method (experiments in 6.2 and 6.3) is appreciated. This is the first paper I have seen that gives convincing evidence of how intervening in the reward mechanism (section 6.3) can occur in *existing* models, when they are simply given the right meta-scaffold.
- The appendix is a trove of examples and qualitative analyses that help to understand the behaviour (although I would suggest that the authors streamline the presentation a little and show a simplified example of how the improver evolves over several iterations in the main text).
- Good writing, no complaints about style and presentation. (The first lines of the introduction already make the reader think!)

**Reasons To Reject:**

- Some parts of the main paper pose as formal but will be unsatisfying to any reader who seeks a formal description of the problem.  The text uses symbols and assumptions that are never defined and stated, hoping the reader will get the right idea. For example, in Section 3:
  - "Formally, let $\Sigma^*$ denote the finite set of text strings": $\Sigma$ was not defined.
  - "Let $L:\Sigma^*\to\Sigma^*$ be a randomized black-box LM...": what does "randomized black-box LM" mean? (It seems like this can just be any stochastic function, that is, function from $\Sigma^*$ to $P(\Sigma^*)$, where $P(X)$ is the space of probability measures over a discrete set $X$.)
  - "$u_{\rm str}\in\Sigma^*$ is a description which may simply be...": what part of this is the formal definition? Is this simply saying "$u_{\rm str}$ is an element of $\Sigma^*$?
  - I am not aware of any formal meaning assignable to the statement "$f$ is a black-box-function", which is used a few times here.
  - The optimization problem alluded to in the very beginning of the paper is never written down.
    - I hoped to find the optimization problem written upon reading the phrase "find an improver program $I$ with high expected utility when used", but only the expected utility itself was defined. It is not even said over what space the optimization is performed (I guess it is over $I$?).
    - Equation (1) is not a definition. How to interpret a mathematical statement that reads "let $x=y$ and ideally ${\rm predicate}(x,y)$"?
  - In Section 4, paragraph "Intuition", the asymptotics are with respect to which variable?
  - Overall, the mixing of examples/informal language and attempts at formal definitions results in a text that provides neither formality nor intuition. I would suggest to keep the main text informal and keep the definitions/math to Appendix A.
- On experiments:
  - The phenomenon discovered in this paper emerges only at very large scales. For better understanding, it would be good to understand:
    - If a similar phenomenon exists at **smaller** scales: perhaps this can be studied with simpler tasks or with partially hardcoded scaffolding, so as to simplify the job of the LM.
    - Some qualitative explanation of why smaller models fail to meta-self-improve (Figure 4).
  - In the other direction, the evaluation could be expanded with experiments on prompt improvement on some more standard language tasks. Conversely, on the tasks currently evaluated, the comparison is only between the base model and different iterations of the proposed improver -- how would other improvement methods (based on fine-tuning or automatic prompt optimization) do here?
  - Can the evaluation of the accuracy shown in Table 1 also be shown over iterations, as in Figure 4, as well as error intervals if multiple instances of of the experiment were run?
  - I could not find many of the experiment details in the paper, in particular, the total compute budget (number of calls to GPT-4, for instance). Did I miss them?

---

> ### Author Rebuttal · Authors · 2024-05-30
>
> Thank you for the thoughtful questions! We’ll be concise given the 2500-character limit.
> > Notational questions
>
> Indeed some descriptions in the problem statement could be further formalized; we’ll remove the word “formal” from Sec 3. We’ll clarify that the set of strings Σ* is over finite alphabet Σ and the optimization maximizes utility *u* over programs *I*. We’ll also define "black-box function" (the system can execute the function but has no implementation information) and "randomized function" (as you noted, a function to the set P(X) of probability distributions over X). In Sec 4, the asymptotics are w.r.t. the budget parameters (for utility, meta-utility, and LM calls). We are open to moving notation to the appendix and defer to the consensus of the review committee and AC.
> > Some qualitative explanation of why smaller models fail
>
> We discuss this in Sec 5.3, but the takeaway is GPT-3.5 makes reasonable proposals, but its implementations are simplistic (e.g., it proposes genetic algorithms but implements them with random string operations vs LM calls).
> > [T]he evaluation could be expanded with experiments on … standard language tasks.
>
> Yes, this would be exciting future work! We tried limiting scope: we anticipate an optimizer that improves language inputs (e.g. rewriting text to sound better) is less likely to be a good code optimizer, so this would be a nontrivial but key follow-up. We’ll discuss this in our revision.
> > [H]ow would other improvement methods (based on fine-tuning or automatic prompt optimization) do here?
>
> In Sec. 5.1, we compare STOP with two alternatives: CoT and greedy iterative improvement. Unfortunately, fine-tuning wasn’t possible for the closed models used. While our focus was on meta-optimization rather than the seed improver, we note any applicable improvement method can be combined with STOP by employing the method as a seed improver.
> > The total compute budget and error intervals for Table 1
>
> In practice, about 3 thousand GPT4 calls were used per iteration per run, and we’re happy to add error bars.
> > [S]how a simplified example of how the improver evolves over several iterations in the main text
>
> Great idea! As improvers often increase in length with iterations (e.g. Fig 2→7) we couldn’t fit this into the main text, but we’re happy to add it to the Appendix.
> > [Study] with simpler tasks or with partially hardcoded scaffolding
>
> Yes, this might also be possible by increasing budget. We’ll add this discussion, thank you!

---

> > ### Author Response · Authors · 2024-06-06
> > **Have we addressed some of your concerns?**
> >
> > Please let us know if we have addressed your concerns or if you have further questions, as the rebuttal period ends soon.

---

### Official Review · Reviewer_wWye · 2024-05-14

**Rating:** 8
**Confidence:** 3
**Ethics Flag:** 1

**Summary:**

This paper introduces Self-Taught Optimizer (STOP), a meta-optimization approach to improve a “scaffolding” program which issues multiple calls to an LLM to search for code solutions to solve downstream tasks. The key idea of the proposed method is to formulate the process of generating the scaffolding program itself as another search problem. Specifically, a scaffolding program takes as input a seed code solution to a downstream task and a utility function that measures solution quality, and issues multiple LLM calls to search for better candidate code solutions to solve the problem. Since the scaffolding program itself is also a code snippet, it could also be used as the input seed code solution to the scaffolding program to improve itself. In this case, the utility function of a scaffolding program is defined as the averaged utility when applying that scaffolding program on different downstream tasks.

**Reasons To Accept:**

* The paper is nicely written. The idea of automatic search and application of optimization strategies to language model optimization and recursive self-improvement is quite novel. An exciting future work is perhaps applying STOP to optimize existing LLM self-improvement methods, such as chain/train-of-thought or self-debugging.

* I really like the generalization experiments in Section 5.2, which shows that the improved improver is transferable to unseen downstream tasks.

**Reasons To Reject:**

* While the downstream tasks used to evaluate STOP (Table 1) are reasonable and fairly challenging, it would be nice to see if the improved improver would achieve better solve rates on more open-domain coding tasks, such as EvalPlus, LiveCodeBench for code generation or other recent benchmarks on debugging (DebugBench).

* The analysis of reward hacking presented in Section 6.3 would benefit from further experiments. The current discussion appears somewhat speculative without quantitative results.

---

> ### Author Rebuttal · Authors · 2024-05-30
>
> Thank you for the encouraging and helpful comments! We would love to find ways to expand STOP to the suggested open-domain coding tasks. A key challenge, which we will discuss in the limitations section, is that STOP requires a meta-utility, and while one might use the benchmark score as metautility, that would both be very expensive and prone to overfit to the benchmark.  We also agree with the point about reward hacking and are open to recommendations for improving the presentation. Unfortunately, reward hacking is somewhat qualitative (as designing a reward that is intentionally hackable may defeat the point).

---

### Decision · Program_Chairs · 2024-07-10

**Decision:**

Accept

**Comment:**

The authors propose a method for improving a 'scaffolding program' (i.e., a program that issues multiple calls to LLMs) that is used to improve code solutions. The scaffolding program can then be applied to improve the scaffolding program itself, and the process can be iterated. The method discovers various interesting scaffolding programs, such as algorithms resembling beam search or genetic algorithms, and these programs lead to improved performance on held out test instances.

The framing of the paper and its methodology are quite creative and timely. All of the reviewers are positive on the paper, with some noting the paper's novelty, several potential applications of the ideas, and the paper's high quality analysis. Reviewers were altogether satisfied with the experiments following the discussion, during which initial comments about testing on additional tasks were determined to be fruitful areas for future work, which I agree with. The authors also provide code to further ease the process of expanding upon the ideas. This paper would make a great addition to the COLM program.